# A Parallel Bicomponent TPU/PI Membrane with Mechanical Strength Enhanced Isotropic Interfaces Used as Polymer Electrolyte for Lithium-Ion Battery

**DOI:** 10.3390/polym11010185

**Published:** 2019-01-21

**Authors:** Ming Cai, Jianwei Zhu, Chaochao Yang, Ruoyang Gao, Chuan Shi, Jinbao Zhao

**Affiliations:** 1College of Physics, Qingdao University, Qingdao 266071, China; 2016020202@qdu.edu.cn (M.C.); withxiaodongxie@163.com (J.Z.); 2College of Textiles & Clothing, Industrial Research Institute of Nonwovens & Technical Textiles, Qingdao University, Qingdao 266071, China; gry206430@126.com; 3College of Chemistry and Chemical Engineering, State Key Lab of Physical Chemistry of Solid Surfaces, Xiamen University, Xiamen 361005, China; chaocyang@163.com (C.Y.); jbzhao@xmu.edu.cn (J.Z.)

**Keywords:** electrospinning, thermal stability, isotropic interfaces, polymer electrolyte, lithium-ion battery

## Abstract

In this work, a side-by-side bicomponent thermoplastic polyurethane/polyimide (TPU/PI) polymer electrolyte prepared with side-by-side electrospinning method is reported for the first time. Symmetrical TPU and PI co-occur on one fiber, and are connected by an interface transition layer formed by the interdiffusion of two solutions. This structure of the as-prepared TPU/PI polymer electrolyte can integrate the advantages of high thermal stable PI and good mechanical strength TPU, and mechanical strength is further increased by those isotropic interface transition layers. Moreover, benefiting from micro-nano pores and the high porosity of the structure, TPU/PI polymer electrolyte presents high electrolyte uptake (665%) and excellent ionic conductivity (5.06 mS·cm^−1^) at room temperature. Compared with PE separator, TPU/PI polymer electrolyte exhibited better electrochemical stability, and using it as the electrolyte and separator, the assembled Li/LiMn_2_O_4_ cell exhibits low inner resistance, stable cyclic and notably high rate performance. Our study indicates that the TPU/PI membrane is a promising polymer electrolyte for high safety lithium-ion batteries.

## 1. Introduction

Due to high mechanical strength, excellent chemical and electrochemical stability, polyethylene (PE) and polypropylene (PP) porous membranes have become the most widely used separator for lithium-ion batteries (LIB) [1]. However, the polyolefin separator will melt below 160 °C, limiting its further use in energy storage systems, especially in electric vehicles (EVs) [2,3,4,5]. In addition, the hydrophobic nature of the polyolefin separator with liquid electrolyte becomes another very major problem in its application. The replacement of polyolefin separators with ceramic coating separators (CCSs) has been recognized as an effective approach to address the above problems [6]. However, as a modified polyolefin separator, thermal stability enhancement of CCS is also restricted by the supporting polyolefin membrane [7,8].

Recently, thermo-stable polymer electrolyte, inorganic electrolyte, and composite electrolyte membranes have attracted increasing attention, as they can be used to solve the safety issues of lithium-ion batteries, including coating, sintering, electrospinning and etc. Methods have been explored in the manufacture of new types of electrolyte membranes [9,10,11,12,13,14]. Among those attempts, electrospinning technology as one of the most effective and simple methods to prepare porous polymer electrolyte is receiving a great deal of attention [15,16,17]. The as-prepared polymer electrolyte tends to have high porosity; thus the cyclic and rate performance of assembled lithium-ion battery will be effectively improved [18,19]. However, traditional electrospinning polymer electrolytes are not entirely able to fulfill the complicated application requirements of batteries [20,21,22].

In this work, we introduced the side-by-side electrospinning method to overcome the safety deficiencies of lithium-ion battery for the first time [23,24]. It should be noted that two non-mixed polymer co-occurr on the same cross section of one fiber prepared with this method. The as-prepared membrane can retain the advantages of using a single component, while having better properties by associating two component. Polyimide (PI) as one of the most well-known thermal stable polymeric materials; it is widely used in high safety batteries. Nevertheless, weak mechanical strength hinders its applications in high safety batteries [25,26,27,28]. Thermoplastic polyurethane (TPU) with strong polar groups shows high elasticity, excellent tensile and aging resistance, and experiments have proved that it can perform well in lithium batteries as a polymer electrolyte, but that its elastic elongation is too high and its thermal stability is low [29,30,31]. Considering the safety requirements in terms of the thermal and mechanical strength of lithium-ion batteries, PI and TPU were selected, and a side-by-side TPU/PI polymer electrolyte was manufactured. As expected from using a side-by-side structure and the chosen basic materials, the as-prepared TPU/PI polymer electrolyte displayed enhanced mechanical strength and excellent thermal stability. Moreover, due to the high porosity of TPU/PI polymer electrolyte, the ether bond of TPU and the affinity to electrolytes of PI [28,31], electrolyte uptake and ionic conductivity of TPU/PI polymer electrolyte were increased significantly, which could guarantee more lithium-ion fast transferring inside the polymer electrolyte, thus resulting in good long term cyclic and rate performance of the lithium-ion batteries [17]. TPU/PI polymer electrolyte with the above advantages was shown to be a promising electrolyte for high safety lithium-ion batteries, and the side-by-side electrospinning method also provides new ideas for the preparation of polymer electrolytes.

## 2. Materials and Methods

### 2.1. Materials

Thermoplastic polyurethanes (TPU) was purchased from BASF (Rhineland-Palatinate, Germany). Polyimide (PI, Mw = 80,000) was from Hangzhou Plastic UNITA special Technology Co., Ltd. (Hangzhou, China). Dimethylformamide (DMF) were obtained from the Sinopharm Chemical Reagent Co., Ltd. (Shanghai, China). Fluorescein (Mw = 332.31) was from Tianjin Dingshengxin Chemical Co., Ltd. (Tianjin, China). Rhodamine B was from Tianjin Kemiou Chemical Reagent Co., Ltd. (Tianjin, China). PE separator with a thickness of 20 μm was purchased from Celgard (Charlotte, NC, USA). All reagents were originally used without further purification.

### 2.2. Preparation of Side-By-Side Fibers

The PI solution (20 wt % in the solvent of DMAC), and the TPU solution prepared by dissolving 20 wt % of TPU in the DMF and stirring for 7 h at room temperature were used for electrospinning process. As Figure 1 shows, the PI and TPU solution were placed into syringes separately, and a Teflon tube was used for holding the two needles. The Teflon tube was extended 5 mm from the tip of the needle so that there would be enough room to form a transition layer after the interdiffusion of two solutions, but also so that the Teflon tube would not be long enough to affect the voltage [32]. A high voltage power supply (GAMMA, Washington DC, USA), two propulsion devices (LSP02-1B, Baoding, China) and a receiver roller (Aluminum foil, Qingdao, China) were constructed as the electrospinning devices. Electrospinning was conducted at 24 kV with an injection rate of 0.5 mL/h, and the distance between syringe and receiving plate was fixed at 15 cm. The experiment was operated at room temperature; the air humidity requirement of the environment is 40 ± 5%. The thickness of polymer electrolyte was controlled by time, and the as-prepared polymer electrolyte was used after drying at 60 °C for 24 h to remove any remaining solvent.

As comparisons, pure TPU and PI membranes were manufactured by the traditional single needle method with the same parameters of the side-by-side electrospinning. The mixed TPU, PI (TPU+PI) membrane was obtained by cross-electrospinning. Four needles were alternately arranged on the electrospinning device; two of them were filled with TPU solution, and the others were for the PI solution. Electrospinning was performed under the conditions described above.

### 2.3. Electrode Preparation and Cell Assembly

Next, 90 wt % LiMn_2_O_4_, 5 wt % super-P, and 5 wt % PVDF were mixed together for the cathode preparation and metal Li was used as the anode. The PE and TPU/PI were used as polymer electrolyte and batteries were assembled in argon gas with a glove box (MIKROUNA, Shanghai, China) after injecting the same volume of electrolyte (Ethylene carbonate (EC): Diethyl carbonate (DEC): Dimethyl carbonate (DMC) 1:1:1 in Volume with 1 Mol·L^−1^ LiPF_6_).

### 2.4. Measurements and Characterization

The morphology of the membranes was observed using a scanning electron microscope (Phenom ProSEM, Shanghai, China) after coating Au layer with a sputter-coater (SBC-12, Beijing KYKY Technology Co., Ltd., Beijing, China). The fluorescence-added bicomponent fibers were identified by fluorescence microscope (OLYMPUS BX51, Tokyo, Japan). Rhodamine B (0.5 wt %) was added to the PI solution, which shows intense red fluorescence, and Fluorescein (0.5 wt %) was put into the TPU solution, which exhibited strong green fluorescence. The fluorescence microscope was used to verify the component of side-by-side bicomponent fiber.

Instron 3300, with a speed of 5 mm min^−1^, was used for the mechanical strength test. The spindle membranes used for mechanical stress test were 40 ± 2 μm in thickness, 1.0 cm in width and 10.0 cm in length. The squared membranes (4 cm^2^) were conducted in a drying oven at different temperatures for heat treatment test. The electrolyte contact angle was measured with a contact angle goniometer (JY-PHb, Chengde Jinhe Instrument Manufacturing Co., Chengde, China).

The porosity of the PE and TPU/PI was calculated as the following equation with the n-butanol uptake method:(1)$$P(\%)=M_{\mathrm{BuOH}}/(\rho_{\mathrm{BuOH}}\times (M_{\mathrm{BuOH}}/\rho_{\mathrm{BuOH}}+M_{m}/\rho_{P}))\times 100\%$$
where *ρ*_BuOH_ and *ρ*_P_ represented the densities of n-butanol and polymer respectively, while M_BuOH_ and M_m_ represented the mass of membrane before and after absorbed n-butanol. The following equation was used for calculating the electrolyte uptake of PE and TPU/PI polymer electrolyte:(2)$$\mathrm{Uptake}\;\%=\frac{W-W_{0}}{W_{0}}\times 100\%$$
where W_0_ and W were the weight of membrane before and after absorbing liquid electrolyte.

PE and TPU/PI polymer electrolyte adsorbed electrolyte and sandwiched between two stainless steel electrodes were used for the linear sweep voltammograms (LSV) and ionic conductivity (electrochemical impedance spectroscopy, EIS) measurement, an electrochemical workstation (3000A-DX, Parstat, NJ, USA) was used. Batteries with PE and TPU/PI membranes before charging were used for the battery resistance test.

The cells assembled with PE and TPU/PI were used for investigating the cycle and rate capability by using an electrochemical test equipment (LAND-V34, Land Electronic, Wuhan, China). The cyclic performance of the batteries was determined by charging to 4.2 V and discharging to 3.0 V at 1.0 C. The rate performances were carried out at current rates of 1.0 C, 5.0 C, 10.0 C, 20.0 C and 1.0 C.

## 3. Results and Discussions

The less uniform of current inner the battery, the easier it is to develop “dendritic” lithium. The inner current is mainly determined by pore distribution of separator. As shown in Figure 2, the PE separator showed a typical oval porous structure and compact pore distribution. This structure was helpful for batteries to reduce the possibility of facing “dendritic” lithium threat [33]. Compared with PE separator, the TPU, PI and TPU/PI membranes formed by the random deposition of electrospinning fibers showed a three-dimensional porous structure. The pore distribution of the membranes were uniform, which is conducive to a uniform distribution of current inner batteries. The pores and porosity of this structure were high, which has advantages and disadvantages: it can improve rate performance of the battery, but the safety of the membrane in terms of e.g., mechanical strength would be reduced greatly [14].

As Figure 2d shows, side-by-side fibers appeared in TPU/PI membranes. In order to distinguish different components in the TPU/PI fibers, fluorescent labeling experiment was conducted. Fibers were selected at a distance of 10 cm from the needle tip to avoid the diameter of the fiber going beyond the distinguishing range of the fluorescence microscope. The result is shown in Figure 2f; green TPU and orange PI were observed in one fiber, and there was a clear interface between different components. This was due to the PI and TPU being almost independent in one fiber except for the interface part, resulting in the PI and TPU fibers forming separately, and being interconnected by a mixed interface part after the solution evaporated. The ratio of TPU and PI in the bicomponent fiber was close to 1:1, which indicated that a side-by-side TPU/PI electrolyte was successfully manufactured. The special side-by-side structure would play a vital role in improving the thermal and mechanical strength of the membrane.

Membranes with high mechanical strength and thermal stability can maintain their original shape, even in extreme conditions, which is a very important property for the safety of lithium-ion batteries. The tensile strength of the membranes was measured and is shown in Figure 3. As shown in Figure 3a, the micro-nano pores, high porosity and nature of PI material together resulted in low mechanical strength of PI membrane, which restricted its application in high safety batteries. TPU is well known as an elastic material; its tensile strength is 6.08 MPa, and elongation at break was 357.7%. Surprisingly, the tensile strength of TPU/PI was 8.85 MPa, which was higher than that of the PI and TPU membrane. This can be explained by the fact that TPU and PI were stretched into fibers in the same Taylor cone; thus, TPU and PI fiber were in a symmetric distribution and interconnected by an interface transition layer formed by the interdiffusion of the TPU and PI solution. Nano-scale interface transition layers were all in the longitudinal direction. According diffusion theory, the mechanical strength of TPU/PI polymer electrolyte was enhanced due to the formation of isotropic interface transition layers [34]. But for the TPU+PI membrane, there was no transition layer and fiber distribution was not uniform. Thus, the membranes would break more easily [35]. Moreover, polymer electrolytes cannot elongate significantly under tension condition in order to avoid contraction in width, and the elasticity of TPU was too high to suit the assembly conditions of lithium-ion batteries. But for TPU/PI polymer electrolyte, the PI would break first due to its low elongation, and TPU would break followed by PI under tension conditions due to the “notch effect”, as shown in Figure 3d. So, the parallel use of PI fibers to decrease the elasticity of TPU fibers is potentially advantageous. Polymer electrolytes with such mechanical and tensile strength could fulfil the requirements for battery assembly.

The main role of the polymer electrolyte is to separate the cathode and anode while keeping the lithium-ion transfer inside the membrane. The battery safety at high temperatures is mainly determined by the polymer electrolyte; the greater the mechanical integrity of membrane at high temperature, the safer of the battery. Thermal treatment can be used to evaluate the thermal stability of membranes. Photographs of PE, TPU and TPU/PI membranes before and after heat treatment for 30 min are presented in Figure 4. The TPU/PI composite membrane showed negligible dimensional change, even at 230 °C for 30 min; meanwhile, the area of the pure TPU membrane reduced to less than 20% after the same treatment. Further probing the effect of heat on the microstructure, the SEM of the membranes after heat treatment were taken. As revealed in Figure 4, PI fiber retained its initial appearance, while melted TPU fibers attached to it at high temperature. The TPU/PI membrane showed no dimensional change, even at 230 °C, since the PI was used as the hot support structure. The thermal safety of the battery would be enhanced when TPU/PI was used as the polymer electrolyte [26].

A hydrophilic membrane can effectively reduce the electrolyte wetting time and improve electrolyte retention capacity [1]. The contact angle of electrolyte on the membrane surface at certain times can reflect the hydrophilicity of the membranes with liquid electrolyte; this was measured and is shown in Figure 5. The contact angle of the PE membrane was about 42 ± 1° while the TPU and TPU/PI membranes were around 0°. The porous structure of the TPU and TPU/PI membrane could rapidly absorb the liquid electrolyte, and the similar ester chemical groups of TPU with the electrolyte further accelerated the wetting speed; thus, the electrolyte contact angel of TPU and TPU/PI membrane were decreased significantly. A polymer electrolyte should be able to absorb and retain electrolyte to guarantee the continued operability of the battery, and membranes with high porosity can retain more electrolyte. The porosity of the TPU/PI membrane was 87.9%, as shown in Table 1, which was twice that of the PE one. The electrolyte uptake of TPU/PI membrane was 665%, which was almost 10 times that of the PE separator, and it was consistent with many previous reports on electrospinning membranes. High electrolyte uptake of the separator could guarantee enough lithium-ion transfer inside the separator, and batteries could have better cyclic and rate performance.

The electrical resistance of the membrane absorbed electrolyte can influence the performance of the battery; this is mainly related to the resistance of liquid electrolytes, the porosity and thickness of membrane, and the tortuosity of pores. Typically, electrochemical impedance spectroscopy (EIS) measurements can accurately reflect the resistance of the membrane. The EIS of PE and TPU/PI membranes in the high frequency portion are shown in Figure 6a. The thickness of the PE and TPU/PI membrane are 20 and 40 μm respectively, and the X-axis intercept of PE and TPU/PI membrane was 0.994 and 0.447 after fitting the line, which indicated that the resistance of TPU/PI polymer electrolyte was less than half that of the PE membrane. Calculated by the formula of R = *ρ*L/S, the ionic conductivity of TPU/PI membrane was 5.06 mS·cm^−1^, which was four times that of the PE separator 1.20 mS·cm^−1^ [17]. Improvements in ionic conductivity were due to the high porosity and excellent electrolyte retention capacity of TPU/PI polymer electrolyte. The battery resistance is consisted to be internal and capacitive resistance, and it is partly determined by the membrane resistance. When assembling the membranes for the batteries, the EIS measurement revealed that the internal resistance of battery with TPU/PI was below 100 Ω, which was much smaller than the 300 Ω of batteries with the PE separator, shown in Figure 6b. The EIS measurement indicated that lithium ions would transfer rapidly inside the TPU/PI polymer electrolyte [36].

Electrochemical stability is a key property for polymer electrolytes vis a vis their practical use in lithium-ion batteries. A linear sweep voltammograms (LSV) experiment in potential range of 0–7.0 V with a scan rate of 1 mV·s^−1^ was performed, and the result is shown in Figure 7a. The PE separator showed obvious decomposition at about 4.4 V. Compared with PE separator, the TPU/PI polymer electrolyte showed obvious decomposition above 5.0 V, which indicated that TPU/PI polymer electrolyte had better electrochemical stability. Further, in order to demonstrate the performance of the batteries with TPU/PI polymer electrolyte, LMn_2_O_4_/(TPU/PI)/Li batteries were assembled and charged-discharged for 100 cycles from 3.0 to 4.2 V at a 1.0 C rate; the results of this test are shown in Figure 7b,c. The traditional LiMn_2_O_4_ battery discharge curves of PE and TPU/PI polymer electrolyte are almost the same, indicating that there are no side effects to the use of the TPU/PI polymer electrolyte. The battery retained 96.6% of its initial capacity with TPU/PI polymer electrolyte after 100 cycles, in contrast to 95.2% for batteries with PE separator, indicating that batteries with TPU/PI polymer electrolyte were better. The improvement of battery performance was associated with the excellent wettability and high electrolyte uptake of TPU/PI polymer electrolyte. For the rate performance measurement, batteries were charged and discharged at rates of 1.0, 5.0, 10.0, 20.0 and 1.0 C. As shown in Figure 7d, the capacity of batteries with PE and TPU/PI membranes are similar at low rates, and decreased quickly with increased rates. Compared with batteries prepared with PE separators, batteries assembled with TPU/PI polymer electrolyte showed significant improvement in capacity retention ability at high rates. The improvement in rate performance of battery may be attributed to porous structure of TPU/PI membrane [37]. Compared with the tunnel structure of PE separator, porous structures could provide more and easier channels for the movement of lithium ions. The unrestricted movement of lithium ions between cathode and anode was beneficial to improve the battery rate performance.

## 4. Conclusions

In summary, a TPU/PI polymer electrolyte was manufactured successfully. Benefiting from its side-by-side structure and basic materials, the as-prepared TPU/PI polymer electrolyte displayed enhanced mechanical strength and excellent thermal stability. Moreover, the electrolyte uptake and ionic conductivity of TPU/PI polymer electrolyte were 10 times and 4 times those of the PE separator respectively. With a TPU/PI membrane as the electrolyte and separator, assembled Li/LiMn_2_O_4_ cells exhibited low inner resistance, stable cyclic performance, and notably high rate performance. The TPU/PI polymer electrolyte with the above advantages is expected to be used in high safety lithium-ion batteries.

## Figures and Tables

**Figure 1 polymers-11-00185-f001:**
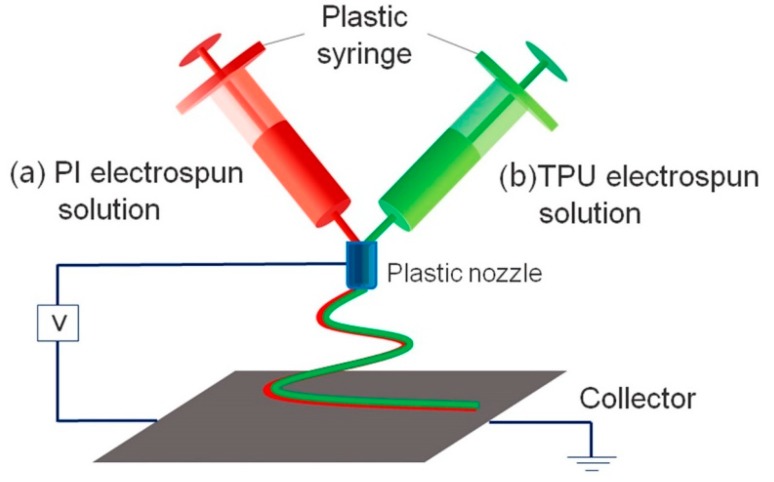
Apparatus schematic of side-by-side electrospinning.

**Figure 2 polymers-11-00185-f002:**
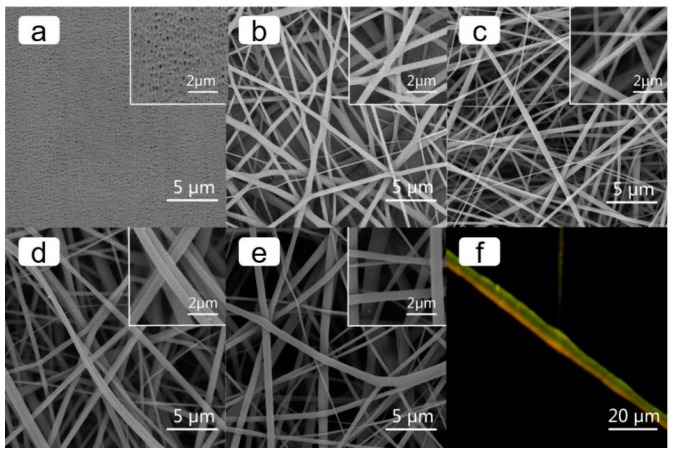
SEM images of (**a**) PE, (**b**) TPU, (**c**) PI, (**d**) TPU/PI and (**e**) TPU+PI membrane. (**f**) Fluorescence microscope images of TPU/PI.

**Figure 3 polymers-11-00185-f003:**
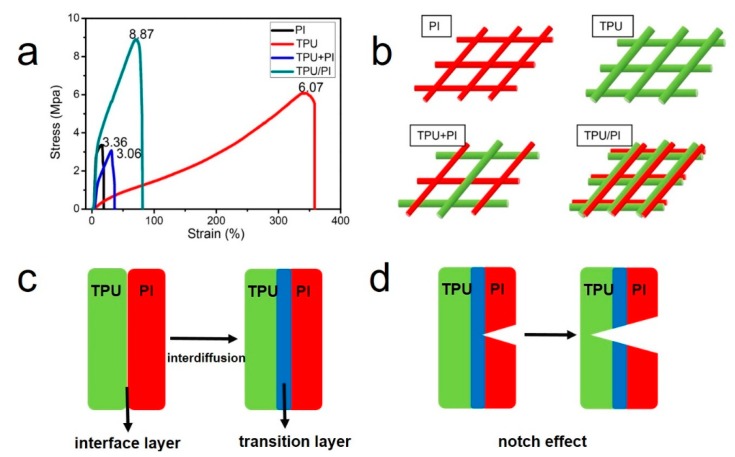
(**a**) fiber stress-strain test curve: TPU, PI, TPU+PI and TPU/PI polymer electrolyte. (**b**) Structure diagram of the fiber membranes. (**c**) Formation diagram of the interface transition layers. (**d**) Schematic diagram of the “notch affect”.

**Figure 4 polymers-11-00185-f004:**
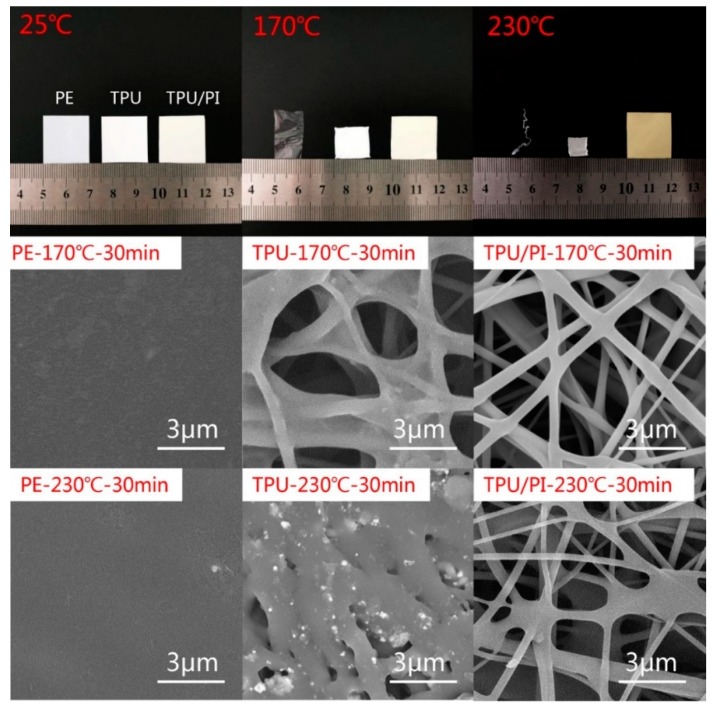
The photograph and SEM of PE, TPU and TPU/PI before and after heat treatment at 170 °C and 230 °C for 30 min.

**Figure 5 polymers-11-00185-f005:**
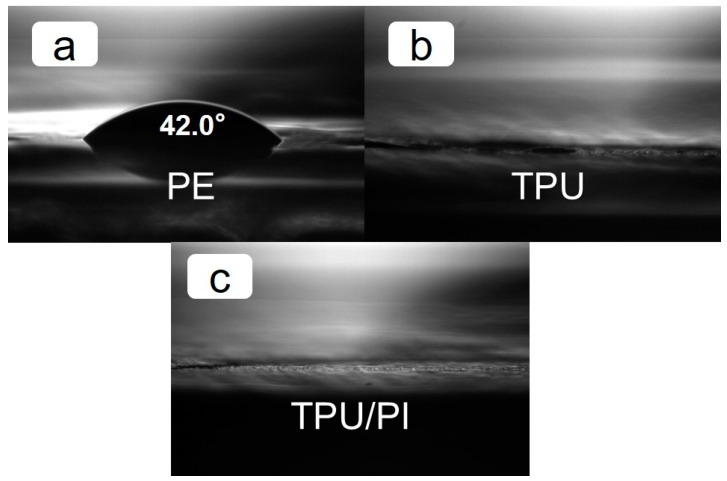
Contact angles of (**a**) PE, (**b**) TPU membrane, (**c**) TPU/PI membrane.

**Figure 6 polymers-11-00185-f006:**
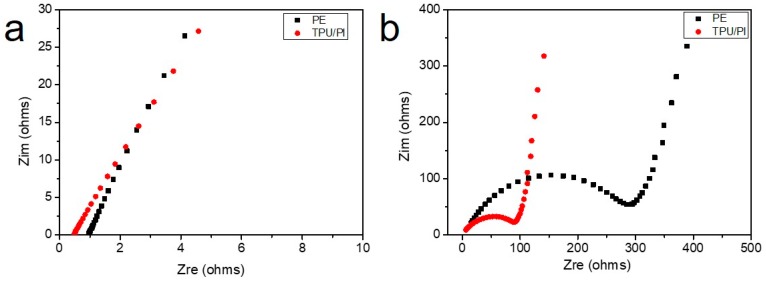
(**a**) EIS of PE and TPU/PI membranes, (**b**) EIS of batteries assembled with PE and TPU/PI membranes.

**Figure 7 polymers-11-00185-f007:**
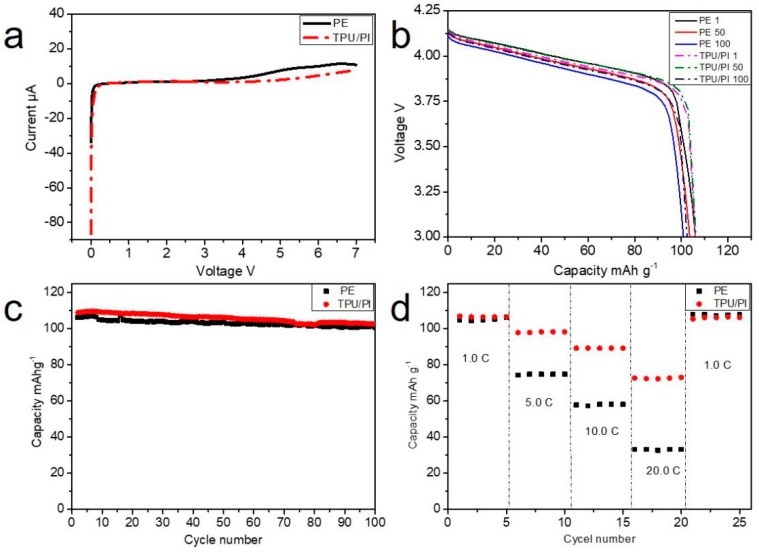
(**a**) LSV of the PE and TPU/PI after saturated with 1 mol·L^−1^ LiPF_6_ electrolyte. (**b**) 1, 50, 100 cycles discharge curves of the batteries with PE and TPU/PI. (**c**) Cyclic and (**d**) rate performance of batteries with PE and TPU/PI polymer electrolyte.

**Table 1 polymers-11-00185-t001:** Physical properties of the membranes.

	PE	TPU/PI
Thickness (μm)	20 ± 1	40 ± 2
Weight (mg)	3.0	2.9
Porosity (%)	42.6 ± 1	87.9 ± 1
Electrolyte uptake (%)	64.5 ± 4	665 ± 6
Ionic conductivity (mS·cm^−1^)	1.20	5.06

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
