# Peer review of "A Parallel Bicomponent TPU/PI Membrane with Mechanical Strength Enhanced Isotropic Interfaces Used as Polymer Electrolyte for Lithium-Ion Battery"

_polymers, 2019, doi:10.3390/polym11010185_

Reviewer 1 Report

The manuscript titled “A parallel bicomponent TPU/PI membrane with mechanical strength enhanced isotropic interfaces used as polymer electrolyte for lithium-ion battery” by Cai et al. details an experimental study on with side-by-side electrospinning method. The thermal stability of PI and mechanical strength of TPU is a key factor here. Its application as polymer electrolyte is also highlighted here. The work is interesting and can be accepted with major revisions. The comments are below –

1. The manuscript could benefit from certain grammatical correction, for instances like “polymer electrolyte is electrochemical stable below 5 V” etc. Such instances are many inside the manuscript and it’s highly recommended that authors revise the manuscript well before submission.

2. Abbreviations like EC, DEC etc. should be clarified before using them directly in main manuscript.

3. Page 4, line 128, current rate of 1.0 C is repeated twice. Should it be something else?

4.  “The special side-by-side structure would play a vital role in improving the thermal and mechanical strength of the membrane”- Authors should provide the rationale behind the statement apart from a stress-strain diagram. Authors should explain the notch effect in view of the tensile experiment. Also, it would be of high interest if authors could provide some examples of battery testing where 100-300% elongation can be seen, which would explain the rationale behind such elaborate tensile testing?

5. Can authors explain the term decomposition in electrochemical stability tests of electrolytes in Fig 7 a, as beyond 5 V the upward trend in PE could also be seen?

6. Fig 7b and 7c don’t actually represent much difference between PE and TPU/PI electrolytes. The major focus was on the wettability of the membranes and porous architecture of latter than former which should enhance its performance than the former. But 1.4% enhancement and 100 cycles do not justify the previous argument. Authors should look more into it and do more cycle tests.

7. Explanation of Fig 7d is poor and justification with a reference number doesn’t soothe readers. Authors should explain it positively, may be from the point of view of channel rich paths etc.

Reviewer 2 Report

The work titled “A parallel bicomponent TPU/PI membrane with mechanical strength enhanced isotropic interfaces used as polymer electrolyte for lithium-ion battery” demonstrates preparation of polyurethane/polyimide membranes by electro spinning technique, where the final fabricated porous structure of the materials resulted into high electrolyte uptake percentage and ionic conductivity respectively. Authors applied electrospinning method aiming to develop material based on the polyimide thermal stable polymer with application of thermoplastic polyurethane. For this reason, authors offered side-by-side method, which results in enhanced performance of the fabricated membranes. Performed work is interesting however missing some critical points.

Figure 4 describes how morphology of the polymer changes after applied annealing step. Can authors quantify those SEM images by measuring and plotting thickness histograms? In addition to this, did authors study changes in mechanical properties before and after annealing on the same material for comparison? From SEM imaged shown on Figure 4, slight thinning effect (i.e. narrowing of fabricated wires) is observed, can author quantify how porosity changes of the fabricated films after thermal treatment? Also, how thermal treatment influences on the capacitance changes of this materials, which is one of the main point of this publication?   

Author Response

Round  2

Reviewer 1 Report

The article can be accepted after some minor correction in grammar.